# The Ability of a Novel Trypsin-like Peptidase Activity Assay Kit to Detect Red-Complex Species

**DOI:** 10.3390/diagnostics12092172

**Published:** 2022-09-08

**Authors:** Michihiko Usui, Masanori Iwasaki, Wataru Ariyoshi, Kaoru Kobayashi, Shingo Kasai, Rieko Yamanaka, Keisuke Nakashima, Tatsuji Nishihara

**Affiliations:** 1Division of Periodontology, Department of Oral Function, Kyushu Dental University, 2-6-1 Manazuru, Kokurakita-ku, Kitakyushu 803-8580, Japan; 2Tokyo Metropolitan Institute of Gerontology, 35-2 Sakae-cho, Itabashi-Ku, Tokyo 173-0015, Japan; 3Division of Infections and Molecular Biology, Department of Health Promotion, Kyushu Dental University, Kitakyushu 803-8580, Japan; 4Division of Periodontal Medicine, Kyushu Dental University, Kitakyushu 803-8580, Japan

**Keywords:** trypsin-like peptidase, periodontitis, red complex

## Abstract

The trypsin-like peptidase activity assay kit measures the trypsin-like protease produced by three red-complex species, *Porphyromonas gingivalis*, *Tannerella forsythia*, and *Treponema denticola*, causing periodontitis, and detects the presence of these bacteria in samples. The purpose of this study was to investigate the relationship between the detection of TLPs by a novel TLP-AA, ADCHECK and the detection of red-complex pathogens by real-time PCR using tongue swabs from patients with periodontitis. The detection limit of trypsin-like protease activity by ADCHECK was validated using the culture supernatants of two different *Porphyromonas gingivalis* bacterial strains. Real-time PCR was performed to determine the number of red-complex species in the tongue coatings of patients with periodontal disease. Trypsin-like protease activity in tongue-swab samples was scored using ADCHECK. ADCHECK successfully detected trypsin-like protease activity in 10^3^* Porphyromonas gingivalis* bacterial strains. The specificity, positive predictive value, negative predictive value, and accuracy of ADCHECK for the presence of red-complex pathogens determined by real-time PCR were 90%, 97%, 98%, and 92%, respectively. ADCHECK is an effective tool for the detection of red-complex pathogens.

## 1. Introduction

Numerous indigenous microorganisms grow and harbor bacteria in the human oral cavity. They create microbial communities on intraoral surfaces, such as the tooth surface, gingival sulcus, and tongue dorsum, with their respective etiological conditions. Periodontitis is a chronic inflammatory disease caused by oral bacteria that induces the production of several inflammatory mediators and cytokines. The progression of periodontitis destroys periodontal tissue, which is composed of the periodontium, cementum, and alveolar bone, which finally leads to tooth loss [1].

*Porphyromonas gingivalis* (*P. gingivalis*), *Tannerella forsythia* (*T. forsythia*), and *Treponema denticola* (*T. denticola*), which are obligatory anaerobic oral bacteria known as ‘red complex’ have been reported to be associated with pathogenesis and progression of periodontitis [2]. These bacteria also produce trypsin-like enzymes that break down proteins, which can hydrolyze the synthetic substrate N-benzoyl-DL-arginine β-naphthylamide (BANA) [3]. The gingipain produced by *P. gingivalis* is one of these trypsin-like enzymes. The BANA test is a simple chair-side test, which detects the putative microorganisms present in plaque, and is widely reported in literature as a reliable measurement of the extent of anaerobic periodontal infection [4,5,6]. Recently, a novel trypsin-like peptidase activity assay kit (TLP-AA), ADCHECK, was developed. This kit is also a rapid and reliable chair-side diagnostic test, which can be performed in approximately 13 min and can provide information about the presence of trypsin-like peptidase (TLP) in samples. There is a difference between the two test kits: the BANA test requires a heating apparatus to increase enzyme activity, whereas the ADCHECK is complete with only a test plate at room temperature. However, the difference in TLP detection capability between ADCHECK and BANA test has been unknown.

Although the tongue dorsum has a unique biofilm of microorganisms, which differs from subgingival periodontal pockets, the relationship between periodontal pathogenic bacteria in the tongue coating and the incidence of periodontal disease remains unclear. Recently, using ADCHECK, we found a correlation between trypsin-like enzyme activity in tongue samples from patients with severe periodontitis and periodontal tissue conditions [7,8]. Real-time PCR is also effective in detecting a small amount of periodontopathogenic bacteria including red-complex bacteria [9,10]. However, it is unclear how many red-complex bacteria, including *P. gingivalis*, *T. forsythia*, and *T. denticola*, can be detected by ADCHECK. In this study, we examined the relationship between the detection of TLP activity by ADCHECK and the detection of red-complex pathogens by real-time PCR, using tongue swabs collected from patients with periodontitis.

## 2. Materials and Methods

### 2.1. Clinical Samples

Tongue-coat samples were collected from patients with periodontitis (*n* = 28) and healthy subjects (*n* = 60) at the Kyushu Dental University Hospital. The sex, mean age, mean probing depth, bleeding on probing (BOP) (%), and number of red-complex pathogens in tongue coating of the patients are shown in Table 1. According to the Centers for Disease Control/American Academy of Periodontology (CDC/AAP), patients with ≥2 interproximal sites with a clinical attachment level (CAL) ≥ 3 mm (not on the same tooth) and ≥2 interproximal sites with a PPD ≥ 4 mm (not on the same tooth) or one interproximal site with a PPD of ≥5 mm were diagnosed with periodontitis [11]. Periodontitis was diagnosed by a periodontist (M.U.). The number of red-complex pathogens in tongue coating was measured by real-time PCR. This study was conducted in full accordance with the ethical principles of the Declaration of Helsinki and was approved by the Ethics Committee of Kyushu Dental University (approval number: 15-6). The periodontitis group showed significantly higher values for age, probing depth, BOP(%), and the number of red-complex pathogens in tongue coat compared to healthy subjects. Samples were collected by swabbing back and forth 10 times across the tongue. Tongue swabs collected from the patients were placed in the extraction buffer (Liquid A) of the ADCHECK kit, and used for each experiment.

### 2.2. TLP-AA Kit

*ADCHECK* (ADTEC Co., Ltd., USA, Japan) is a novel kit used for detecting TLP activity. The mechanism of TLP activity determined using the ADCHECK kit is shown in Figure 1a. BANA substrates are degraded by TLP from bacteria and β-naphthylamide is released. When the released β-naphthylamide is stained with a colorant solution containing 4-dimethylamino cinnamaldehyde, it turns pink. The ADCHECK kit consists of a test-plate disk including BANA substrate, liquid A: buffer to extract the enzyme from bacteria, liquid B: colorant solution to stain β-naphthylamide, and swab to collect samples (Figure 1B). Tongue swabs were collected from the patients (Figure 1c), placed in the extraction buffer of the ADCHECK kit, and stirred (Figure 1d). The samples, which contained BANA as a matrix, were placed on a test-plate disk for 5 s and allowed to react at room temperature (1–30 °C) for 10 min (Figure. 1E). In the experiment with cultured bacteria, 80 μL of cultured bacterial supernatant was placed on a test-plate disk and reacted under the same conditions. If TLP was present in the samples, the enzymatic activity of the relevant enzyme in the samples was expected to disintegrate the matrix and subsequently release β-naphthylamide. After 10 min, one drop of the 4-(dimethylamino) cinnamaldehyde colorant was dropped onto the test-plate disk (Figure 1f), and the released β-naphthylamide reacted to produce a pink compound. Three minutes after application of the chromophore, the color intensity of the matrix disk was visually evaluated using a sample for score interpretation (Figure 1g). The color intensity of the matrix disks was ranked on a scale ranging from 1 to 5, with a stronger pink color (higher score) indicating intense trypsin-like peptidase activity. The scoring system was as follows: (1) equivalent to ≥10 units/mL trypsin, (2) equivalent to ≥25 units/mL, (3) equivalent to ≥100 units/mL, (4) equivalent to ≥500 units/mL, and (5) equivalent to ≥5000 units/mL. One unit was defined as the amount of enzyme required to produce a ΔA253 of 0.001 per minute with Nα-benzoyl-l-arginine ethyl ester as the substrate, at pH 7.6 and 25 °C.

The BANA test (BANAMET LLC, Ann Arbor, MI, USA) is a popular TLP-AA kit that has been used for many years. The BANA test was performed according to the manufacturer’s protocol as described by Loesche et al. [4,6]. First, 80 μL of supernatant from cultured *P. gingivalis* bacterial strains was applied to a cellulose strip in the lower half of a BANA card. Then, the upper strip containing the Fast Black dye was moistened with distilled water to activate it. The lower portion of the card was bent forward such that the two strips were in contact. The card was placed in a heating apparatus (BANA processor) with the two strips in contact. The card was incubated at 55 °C for 15 min, after which the upper strip was examined for the presence of blue spots.

### 2.3. Bacterial Cultures

*Porphyromonas gingivalis* JCM 12257 and ATCC BAA-308 were purchased from RIKEN BRC (Tsukuba, Japan). These were maintained at 37 °C in enriched brain heart infusion broth (BHI) containing BHI (37 g/L, Oxoid, Hampshire, UK) supplemented with yeast extract (5 g/L), hemin (5 mg/L), vitamin K1 (1 mg/L), and cysteine (1 g/L), under anaerobic conditions using an anaerobic glove box (gas phase, 10% CO_2_/10% H_2_/80% N_2_, Model ANX-3; Hirasawa Co., Tokyo, Japan) or AnaeroPack system (Mitsubishi Gas Chemical Co., Tokyo, Japan). These bacterial strains were pre-cultured for 24 h, and then, incubated for another 48 h. Serial dilutions of the collected bacteria were made and colony counts were performed on plain medium. The culture supernatant was used for ADCHECK and BANA tests. In the cross-intersectionality test, 40 oral bacteria other than red-complex bacteria were cultured under their respective suitable culture conditions, and the culture supernatant was appliqued to the ADCHECK test plate. Culture methods for these 40 bacterial species are described in the Appendix A.

### 2.4. Real-Time PCR

Counts of the three bacterial species were determined for each sample using real-time PCR. Genomic DNA (gDNA) was isolated from extraction-buffer-dissolved tongue-coat samples using the QIAamp DNA Mini Kit (Qiagen, Hilden, Germany), according to the manufacturer’s recommendations. Amplification and detection of DNA with species-specific primers by real-time PCR was performed using the StepOne Real-Time PCR System (Applied Biosystems). For each RT-PCR, universal SYBR Green Supermix (Bio-Rad Laboratories, Hercules, CA, USA) and a 20 μL total PCR amplification volume for each reaction were used. The DNA amplification conditions for PCR with species-specific primers for *P. gingivalis*, *T. forsythia*, and *T. denticola* were 30 s initial denaturation at 95 °C, followed by 50 consecutive cycles at 95 °C for 10 s, 60 °C for 20 s, and 72 °C for 20 s, for data collection. The number of each bacterium was calculated using a calibration curve. The forward and reverse species-specific primer sequences used were as follows:

*P. gingivalis*; F: CCGCATACACTTGTATTATTGCATGATATT, R: AAGAAGTTTACAATCCTTAGGACTGTCT, *T. forsythia*; F: ATCCTGGCTCAGGATGAACG, R: TACGCATRCCCATCCGCAA *T. denticola*; F: CCTTGAACAAAAACCGGAAA, R: GGGAAAAGCAGGAAGCATAA.

Primer designs were obtained from Sakamoto et al. [12].

### 2.5. Statistical Analysis

The Mann–Whitney U-test was used to determine the significance between healthy and periodontitis groups. *p*-values < 0.05 were considered to be statistically significant. The frequencies of ADCHECK results according to different bacterial levels determined by real-time PCR were analyzed using the chi-square test. The sensitivity, specificity, positive predictive value (PPV), negative predictive value (NPV), and accuracy values of the ADCHECK related to real-time PCR were also determined using the chi-square test. The significance level was set at 5% (*p* < 0.05). McNemar’s test was also performed and the kappa coefficient was calculated. The sample size was determined according to [13]. Considering the estimated prevalence of a positive result of real-time PCR, the red-complex bacterial count of >1000 colony forming units (cfu)/mL was around 30% (28/88), the sample size was estimated at 67 with a sensitivity of ADCHECK = 80%, corresponding to an alpha = 0.05, and power = 80%. The Kruskal–Wallis test and Pearson product-moment correlation coefficient were used to determine significance and correlation in the comparison of the number of red-complex bacteria per ADCHECK score value. *p*-value < 0.05 was considered to be statistically significant.

## 3. Results

### 3.1. Detection of TLPs from P. gingivalis Bacterial Strains by a Novel TLP-AA Kit

First, we examined whether TLPs in the culture supernatant of bacterial strains could be measured by ADCHECK using the *P. gingivalis* bacterial strains. Since the BANA test is a TLP-AA kit that has long been used in clinical practice, we used it as a positive control. Therefore, we compared both ADCHECK and BANA test for TLP detection. *P. gingivalis* bacterial strains JCM12257 and BAA-308 were cultured for 72 h, and the culture supernatant was applied to ADCHECK. The culture supernatant from the culture of 8.0 × 10^2^ cells of JCM12257 cells failed to produce pink coloration on the test plate, but the test plate to which 8.0 × 10^3^ cells of JCM12257 culture supernatants were added stained light pink, suggesting the presence of TLPs. As the number of bacteria in JCM12257 increased, the pink color of the test plate became more intense. On the other hand, the BANA test detected BANA-degrading enzyme activity in the culture of 8.0 × 10^6^ cells of JCM12257 cells (Figure 2a). Similar to JCM12257, plate disks applied the culture supernatant of 10^3^ BAA-308 cells stained pink. As in JCM12257, the pink color of the plate disk became darker depending on the number of BAA-308. Interestingly, the BANA test failed to detect the BANA-degrading enzymes in the culture supernatant (Figure 2b). These data indicated that ADCKECK is more sensitive than the BANA test. These data also suggested that *ADCHECK* could detect TLP in the culture supernatant of *P. gingivalis* bacterial strains.

To verify whether ADCHECK detects only TLPs of the red-complex species, a crossover examination with other oral bacteria was performed. Although, as shown in Table 2, culture supernatants of 40 different oral bacteria other than red-complex bacteria were applied to ADCHECK, TLPs in these species were not detected (Table 2). These results suggest that ADCHECK does not detect enzymes other than TLPs produced by bacteria, other than the red-complex bacteria.

### 3.2. Comparison of Real-Time PCR and a Novel TLP-AA Kit in Clinical Samples

Next, we compared the detection of red complexes by real-time PCR and TLP activity measurement by ADCHECK using tongue-coat samples from periodontitis patients and healthy individuals. In the real-time PCR method, a positive result was obtained when the number of red-complex bacteria was more than 1000 colony forming units (cfu). For detection by ADCHECK, a score of 2 or higher was considered positive. Based on these criteria, the sensitivity, specificity, PPV, NPV, and accuracy of the ADCHECK for the presence of red-complex pathogens determined by real-time PCR were examined. In this experiment, the specificity, PPV, NPV, and accuracy of ADCHECK compared to the real-time PCR method were 98%, 97%, 90%, and 92%, respectively, all of which were higher than 90%. However, the sensitivity was slightly lower at 83% (Table 3). The Kappa coefficient of these tests was also 0.83, higher than 0.8, indicating high reproducibility of real-time PCR and ADCHECK. These data suggest that ADCHECK can detect red-complex pathogens in clinical samples and has high specificity and NPV for real-time PCR.

In addition, the relationship between bacterial number of red-complex pathogens by RT-PCR and ADCHECK score values was verified using tongue-coat samples. The number of red-complex pathogens at an ADCHECK score value of 2 was significantly higher compared with ADCHEK score value of 1. On the other hand, the number of red-complex bacteria at ADCHECK score value of 3 was higher than at an ADCHECK score value of 2, however, the difference was not significant (Figure 3a). The correlation between the number of red-complex bacteria and the ADCHECK score value was further investigated. The r value was 0.580 (*p* < 0.05), and there was a correlation between the number of red-complex bacteria and the ADCHECK score value (Figure 3b).

## 4. Discussion

In this study, we found that TLP activity produced by *P. gingivalis* cultured bacterial strains could be detected by a novel TLP-AA kit, ADCHECK, and that the color reaction in ADCHECK progressed with increasing bacterial numbers. In addition, ADCHECK was able to detect TLPs present in the patient samples and showed high accuracy compared to the results obtained by real-time PCR. A positive correlation was also observed between ADCHECK score values and the number of red-complex bacteria. ADCHECK is a kit for easy detection of TLPs by some bacteria species including the red complex. The ADCHECK kit consists of a test plate, lysis solution, and staining solution, and the total time required for detection is less (13 min). Furthermore, ADCHECK does not require an incubator to activate the enzymes produced by the bacteria; therefore, the test can be completed at the chairside. ADCHECK is different from the BANA test, a similar TLP-AA kit, in that it does not require an incubator. While it has been known for some time that there is a relationship between the presence of periodontal bacteria in periodontal pockets and the progression of periodontal disease [11], we recently found a correlation between the amount of TLPs on the tongue coat and severe periodontitis [8]. Even in an environment such as a physical examination where there is no dentist or dental hygienist to measure periodontal pocket depth, the red-complex pathogens in the tongue coat can be easily and precisely monitored using ADCHECK, which has a detection capacity similar to that of real-time PCR. The specificity of ADCHECK for real-time PCR was very high at 98%. The high specificity is suitable for screening of diseases. These suggest that ADCHECK may be a useful tool for screening of periodontitis.

A positive correlation was observed between ADCHECK score values and the number of red-complex bacteria (Figure 3b), and the number of red-complex pathogens. In ADCHECK, score 2 was significantly increased compared to score 1. However, there was no significant difference between the number of red-complex bacteria in ADCHECK score 3 and score 2 (Figure 3a). One reason for this may be that the number of specimens showing score 3 was small. Another reason is the presence of other bacteria producing TLPs. In addition to the red-complex bacteria, other bacteria producing TLPs are present in the oral cavity, for example, some bacteria of the *Capnocytophaga* species [14]. ADCHECK’s detection of TLPs produced by these bacteria may have led to the differences in the number of red-complexed bacteria by RT-PCR among the score values. In the future, we plan to increase the number of clinical samples to re-examine the relationship between the ADCHECK score value and the number of red-complex pathogens, as well as to validate the supernatants of cultured bacterial strains producing TLPs other than red-complex bacteria using ADCHECK.

The BANA test is another previously developed method to detect BANA-degrading enzymes produced by red-complex bacteria. There have been many reports of the detection of red complexes using the BANA test. Andrade et al. reported the ability of the BANA test to detect red-complex pathogens compared with checkerboard DNA–DNA hybridization. The sensitivity of the BANA test to DNA is extremely high (96%) [15]. On the other hand, as shown in Table 3, the sensitivity of ADCHECK for real-time PCR was 83%, which is slightly lower. This difference may be due to the difference in the cutoff values of checkerboard DNA–DNA hybridization and real-time PCR. In Andrade et al.’s study, the cutoff value for checkerboard DNA–DNA hybridization was set at 10^4^ cells, whereas in our study, the cutoff value for real-time PCR was set at 10^3^ cells. If we raise the cut-off value of our real-time PCR to 10^4^, the sensitivity of ADCHECK may also increase. In addition, the specificity of the BANA test for the checkerboard DNA–DNA hybridization was extremely low at 12%, while the specificity of the ADCHECK for the real-time PCR method was extremely high (98%). These data suggest that ADCHECK is a test kit that can detect red-complex bacteria stably and with high accuracy. The detailed ability differences between the test kits will be verified in the future.

*P*. *gingivalis*, one of the red-complex bacteria, is thought to be the causative agent of periodontal disease because of its strong pathogenic properties, such as gingipain and LPS [16]. Recently however, the keystone pathogen hypothesis has emerged as a possible mechanism for the development of periodontal diseases. It is hypothesized that by manipulating the innate immune system and disrupting leukocyte function, *P. gingivalis* not only promotes its own survival and amplification, but also that of the entire bacterial community. Contrary to the majority of bacterial species, which affect inflammation when present in large numbers, keystone pathogens can induce inflammation even when present in small numbers [17,18]. In other words, detecting the presence of a small amount of *P. gingivalis* and removing it is important for managing periodontal tissue against periodontitis. Based on this hypothesis, tests that can detect even a small amount of *P. gingivalis* will become important in the future, and, in this regard, ADCHECK will be a useful tool in future periodontal therapy, as it can detect smaller amounts of red-complex bacteria than the BANA test.

The gold standard for the examination of periodontitis is the measurement of probing pocket depth and clinical attachment level (CAL), and the evaluation of alveolar bone by radiographic imaging. Although bleeding on probing (BOP) is a classical and beneficial indicator of periodontitis progression [19,20], it cannot distinguish between minute and massive bleeding and is not quantitative. Recently, inflammatory cytokines in gingival crevicular fluid (GCF), such as MMP-8, IL-1, and TNF-α, were measured and found to be associated with disease progression and response to treatment [21,22,23]. However, the measurement of these cytokines is performed by ELISA, which is expensive in terms of purchasing equipment and testing costs. Periodontal bacteria are the cause of periodontitis, and the three red-complex species in particular are directly involved in the progression of periodontitis. Real-time PCR is the most sensitive method for detecting periodontal bacteria. However, it requires large and expensive PCR equipment, and the reagents are expensive, resulting in high testing costs. The novel TLP-AA kit, ADCHECK, used in this study is cheaper than ELISA and uses real-time PCR methods. In addition, the detection time of this kit is only 13 min, which is extremely small compared to that of ELISA and real-time PCR methods. These results suggest that this kit is a cheaper and highly accurate tool that can perform easily at the chairside and will also help in periodontal treatment by providing information on red-complex pathogens in the oral cavity.

## 5. Conclusions

In this study, ADCHECK succeeded in detecting TLPs on the tongue coat with high accuracy, suggesting that ADCHECK is a more effective tool for monitoring TLPs produced by bacteria including the red-complex pathogens. In the future, if ADCHECK can detect TLPs in periodontal pockets, it may be possible to determine the effectiveness of treatment for periodontitis, i.e., whether periodontopathic bacteria have been removed.

## Figures and Tables

**Figure 1 diagnostics-12-02172-f001:**
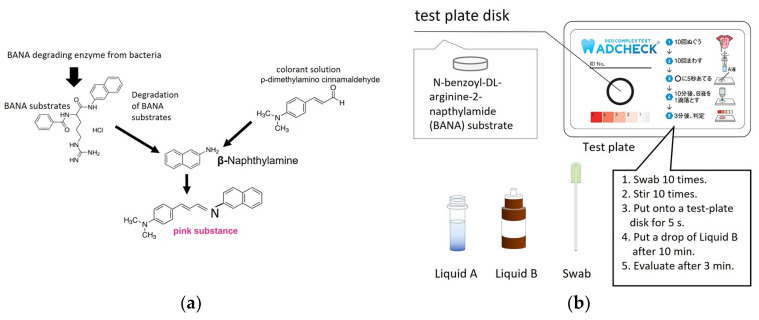
Procedure for using a novel TLP-AA kit, ADCHECK. (**a**) The mechanism of ADCHECK to detect TLP activity. (**b**) Components of the ADCHECK kit. The size of the test plate is 55 mm in length and 85 mm in width. This kit also includes Liquid A for extraction TLP, Liquid B for color reaction, and a swab for collecting the sample. (**c**) Collecting a sample (tongue coating) with a swab. (**d**) Extraction of TLP from samples. The swab containing the sample was stirred well in the tube containing Liquid A. (**e**) The sample was applied to the test plate. The samples were put onto a test-plate disk for 5 s and allowed to react at room temperature for 10 min. (**f**) One drop of 4-dimethylamino cinnamaldehyde colorant (Liquid B) was dropped onto the test-plate disk. (**g**) Color reaction to pink to the released β-naphthylamide. Three minutes after the application of Liquid B, the intensity of the color of the plate disk was visually evaluated using color chart.

**Figure 2 diagnostics-12-02172-f002:**
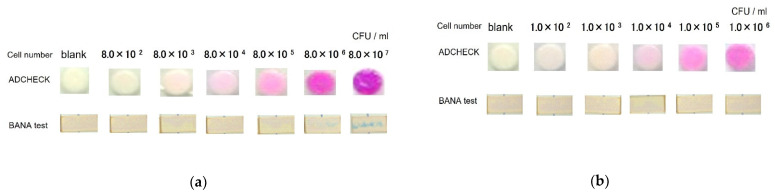
Detection of TLPs from cultured *P. gingivalis* bacterial strains by TLP-AA kit. The cell supernatants of each number of JCM12257 (**a**) and BAA-308 (**b**) in culture were examined by ADCHECK and BANA test. In the ADCHECK test, pink coloration was diagnosed as positive for TLP, while in the BANA test, the appearance of blue lines was diagnosed as positive for TLP.

**Figure 3 diagnostics-12-02172-f003:**
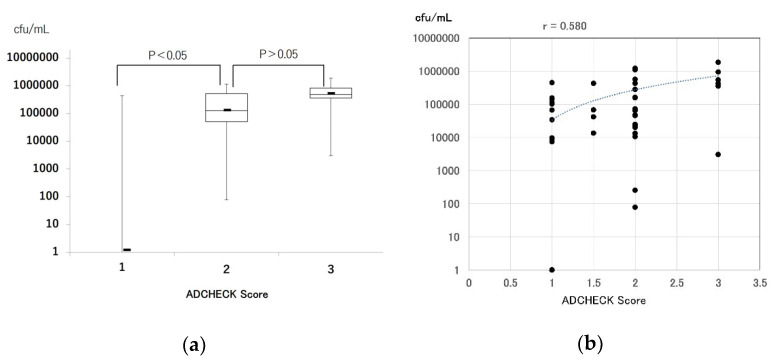
Comparison of the number of red-complex bacteria determined by real-time PCR per ADCHECK score value in tongue-coat samples. (**a**) bar graph and (**b**) scatter plot.

**Table 1 diagnostics-12-02172-t001:** Patient characteristics.

	Sex	Age (Years) Mean (SD)	Probing Depth (mm) Mean (SD)	BOP (%) Mean (SD)	Number of Red Complex Pathogens in Tongue Coat Mean (SD)
Healthy (60)	Male: 12	26.9 (5.2)	1.4 (0.3)	2.8 (2.0)	27,439 (82,438)
	Female: 16				
Periodontitis (28)	Male: 19	52.1 (11.3) *	2.3 (1.3) *	8.2 (5.3) *	405,243 * (506,060)
	Female: 41				

* *p* < 0.05 red-complex pathogens: *P. gingivalis*, *T. forsythia*, and *T. denticola*.

**Table 2 diagnostics-12-02172-t002:** Oral bacteria using crossover examination of ADCHECK.

	Bacterial Name	Strain No.	cfu/mL		Bacterial Name	Strain No.	cfu/mL
1	*Bordetella pertussis*	NBRC107857	1.69 × 10^8^	21	*Streptococcus mutans*	NBRC13955	1.13 × 10^8^
2	*Candida albicans*	NBRC1385	5.17 × 10^8^	22	*Streptococcus oralis*	JCM12997	8.60 × 10^7^
3	*Corynebacterium diphtheriae*	JCM1310	5.25 × 10^6^	23	*Streptococcus pneumoniae*	NBRC102642	6.45 × 10^9^
4	*Enterococcus durans*	NBRC100479	3.75 × 10^8^	24	*Streptococcus pyrogenes*	JCM5674	1.07 × 10^8^
5	*Enterococcus faecalis*	NBRC100480	6.00 × 10^8^	25	*Streptococcus intermedius*	ATCC9895	6.50 × 10^8^
6	*Haemophilus infuluenzae*	ATCC9006	1.56 × 10^8^	26	*Streptococcus sanguis*	JCM5708	2.85 × 10^7^
7	*Listeria monocytogenes*	JCM7671	1.07 × 10^9^	27	*Neisseria gonorrhoeae*	ATCC19424	1.02 × 10^9^
8	*Moraxella catarrhalis*	ATCC8176	2.14 × 10^8^	28	*Neisseria meningitidis*	ATCC13077	4.20 × 10^9^
9	*Mycoplasma orale*	NBRC14477	2.48 × 10^7^	29	*Neisseria sicca*	ATCC9913	6.20 × 10^8^
10	*Mycoplasma pneumoniae*	NBRC14401	1.81 × 10^7^	30	*Neisseria subflava*	ATCC19243	2.11 × 10^8^
11	*Mycoplasma salivarium*	NBRC14478	1.02 × 10^6^	31	*Streptococcus pyogenes*	ATCC12353(T12)	2.35 × 10^7^
12	*Mycoplasma hominis*	NBRC14850	1.34 × 10^5^	32	*Streptococcus pyogenes*	ATCC12962(T28)	4.00 × 10^6^
13	*Proteus vulgaris*	NBRC3045	4.25 × 10^9^	33	*Streptococcus pyogenes*	BAA-1066(M4)	2.30 × 10^7^
14	*Pseudomonas aeruginosa*	NBRC12689	3.85 × 10^9^	34	*Escherichia coli*	ATCC11775 (JCM1649)	2.40 × 10^9^
15	*Serratia marcescens*	NBRC3046	1.04 × 10^9^	35	*Klebsiella pneumoniae*	ATCC13883 (JCM1662)	1.76 × 10^8^
16	*Staphylococcus aureus*	NBRC102135	3.17 × 10^9^	36	*Citrobacter freundii*	JCM1657	2.24 × 10^9^
17	*Staphylococcus epidermidis*	NBRC100911	1.38 × 10^8^	37	*Salmonella enteritidis*	IFO3313	1.15 × 10^9^
18	*Streptococcus agalactiae*	JCM5671	1.70 × 10^8^	38	*Salmonella typhimurium*	IFO13245	8.2 × 10^9^
19	*Streptococcus anginosus*	JCM12993	2.08 × 10^8^	39	*Streptococcus dysgalactiae* *subsp. equisimilis*	ATCC12388	3.7 × 10^8^
20	*Streptococcus dysgalactiae* *subsp. dysgalactiae*	JCM5673	4.04 × 10^8^	40	*Streptococcus constellatus* *subsp. constellatus*	JCM12994	5.05 × 10^8^

**Table 3 diagnostics-12-02172-t003:** Comparison of ADCHECK and real-time PCR in clinical samples.

		Real-Time PCR																
		*P. gingivalis*			*T. forsythia*			*T. denticola*			Red Complex			Sensitivity	Specificity	PPV	NPV	Accuracy
		POS	NEG	SUM	POS	NEG	SUM	POS	NEG	SUM	POS	NEG	SUM					
ADCHECK	POS	15	16	31	29	2	31	13	18	31	30	1	31	83%	98%	97%	90%	92%
	NEG	9	48	57	9	48	57	3	54	57	6	51	57					
	SUM	24	64	88	38	50	88	16	72	88	36	52	88					

POS, positive; NEG, negative; PPV, positive predictive value; NPV, negative predictive value *p* < 0.05 (chi-squared test). Kappa coefficient: 0.83.

## Data Availability

Data supporting the findings of this study are available from the corresponding author, M.U., upon reasonable request.

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
