# Peer review of "The Ability of a Novel Trypsin-like Peptidase Activity Assay Kit to Detect Red-Complex Species"

_diagnostics, 2022, doi:10.3390/diagnostics12092172_

Round 1
Reviewer 1 Report
This study investigated the ability of a novel trypsin-like peptidase (TLP) activity assay kit to detect so-called “red complex bacteria”. Tongue swab samples were collected from patients with periodontitis and healthy subjects, and the presence/absence of red complex bacteria, and TLP levels were examined. The TLP activity (≥10 units/mL) was associated with the presence of red complex bacteria (at least one of three red complex species was detected in the sample). The authors concluded that a novel TLP activity assay kit is a more effective tool for the detection of red complex pathogens.
This type of manuscript might be of interest to the readers of Diagnostics. However, there are some concerns in the manuscript, and considerable revision is needed.
Major comments;
- The authors should clarify the novelty of this study. As the authors mentioned, “the BANA test is a simple chair-side test, which detects the putative microorganisms present in plaque, and is widely reported in the literature as a reliable measurement of the extent of anaerobic periodontal infection (Line 43-46)”. Although the novel product presented in the study may be able to diagnose the presence of red complex bacteria with high sensitivity and specificity, it is not a novel. This reviewer also requests to revise the Discussion according to the objective and results of this study.
- The authors mentioned in the Introduction that “we found a correlation between trypsin-like enzyme activity in tongue samples … using ADCHECK” and "it is unclear how many red complex bacteria, including P. g, T. f, and T. d, can be detected by ADCHECK (Line 56-57)”. If so, the relationship between TLP activity and the numbers of each red complex bacterium should be examined using both cultured bacteria and tongue swab samples and compared to the results.
- It is emphasized the importance of P.g as a keystone pathogen and the usefulness of ADCHECK in Discussion (Lines 223-243). While ADCHECK is a kit to know the total TLP activity level of a clinical sample, it does not necessarily correlate with the amount of P.g in clinical sample. The sentences written about the usefulness of ADCHECK should be reconsidered and revised.
Minor comments;
- Page 2, Line 63; It would be helpful for readers to show clinical data (age, gender, mean PD/CAL, mean percentage of sites with BOP positive, plaque score, number of remaining teeth, etc.) of participants as a table.
- Page 2, Line 65; Please provide the reference describing the CDC/AAP criteria for periodontal disease.
- Page 2, Line 71; Please describe the details of the sampling method. Was the sampling method standardized?
- Page 2, Line 97; It is unclear why (purpose) and how (procedure) the BANA test was conducted in this study. The authors should describe the details together with the interpretation of the result.
- Page 3, Line 99; Please correct the typo (Name “Loesche”).
- Page 4, Line 131; The authors mentioned that "Genomic DNA was isolated from cultured bacterial cells using the QIAamp DNA Mini Kit". Did you also isolate the bacterial DNA from tongue swab samples? Please add a description how you process the clinical samples for PCR analysis.
- Page 4, Line 145; Please replace reference 11 with the correct one.
- Page 4, Line 153; Considering the objective of the present study, the TLP activity level of cultured T. f and T. d should be measured.
- Page 6, Line 201; The authors mentioned, "ADCHECK was able to detect the red complex pathogens present in the patient samples” and “the ADCHECK is a test kit that can detect red-complex bacteria stably and with high accuracy (Line 228)”. These sentences do not accurately represent the results and should be revised.
- Page 7, Line 211; The authors mentioned that "the ADCHECK is a useful tool for the treatment of periodontitis”. This sentence is unclear, and please explain in detail the benefit to use ADCHECK for periodontal treatments.
Author Response
Reviewer 1
We wish to express our strong appreciation to the reviewer for their insightful comments on our paper. We feel the comments have helped us significantly improve the paper. We have revised our manuscript as much as possible in accordance with your suggestions. The followings are our responses for your comments.
Major comments;
1.- The authors should clarify the novelty of this study. As the authors mentioned, “the BANA test is a simple chair-side test, which detects the putative microorganisms present in plaque, and is widely reported in the literature as a reliable measurement of the extent of anaerobic periodontal infection (Line 43-46)”. Although the novel product presented in the study may be able to diagnose the presence of red complex bacteria with high sensitivity and specificity, it is not a novel. This reviewer also requests to revise the Discussion according to the objective and results of this study.
Thank you for your comment.
ADCHECK is novel compared to existing product (BANA test) in that it does not require an incubator to activate Trypsin-like protease produced by the red complex and can be reacted at room temperature. This point is described in the introduction to clarify the purpose of this study (L52-55).
2.- The authors mentioned in the Introduction that “we found a correlation between trypsin-like enzyme activity in tongue samples … using ADCHECK” and "it is unclear how many red complex bacteria, including P. g, T. f, and T. d, can be detected by ADCHECK (Line 56-57)”. If so, the relationship between TLP activity and the numbers of each red complex bacterium should be examined using both cultured bacteria and tongue swab samples and compared to the results.
Thank you for your very important point.
We examined the relationship between the number of Red Complex pathogens and the ADCHECK score value using tongue coat samples. The results are shown in Figure 3 (P7).
3.- It is emphasized the importance of P.g as a keystone pathogen and the usefulness of ADCHECK in Discussion (Lines 223-243). While ADCHECK is a kit to know the total TLP activity level of a clinical sample, it does not necessarily correlate with the amount of P.g in clinical sample. The sentences written about the usefulness of ADCHECK should be reconsidered and revised.
Thank you for your valuable comment.
As describe above, we have performed additional experiments and added new figure (Fig 3) regarding the comparison of number of red complex bacteria and TLPs activity produced by these bacteria in clinical samples. We have also revised our DISCUSSION part based on the fact that ADCHECK is a kit that measures TLPs produced by red complex bacteria.
Minor comments;
- - Page 2, Line 63; It would be helpful for readers to show clinical data (age, gender, mean PD/CAL, mean percentage of sites with BOP positive, plaque score, number of remaining teeth, etc.) of participants as a table.
We added clinical data of healthy and periodontitis patients who provided tongue coat samples (P2, Table 1).
- - Page 2, Line 65; Please provide the reference describing the CDC/AAP criteria for periodontal disease.
We added a reference describing the CDC/AAP criteria for periodontal disease as ref. 11.
- - Page 2, Line 71; Please describe the details of the sampling method. Was the sampling method standardized?
We described a method for sampling tongue coat in Materials and Methods part ( L80-82).
- - Page 2, Line 97; It is unclear why (purpose) and how (procedure) the BANA test was conducted in this study. The authors should describe the details together with the interpretation of the result.
As mentioned above (response to major comment 1), the BANA test and ADCHECK differ in whether or not they require an incubator. Since the BANA test is a TLPs measurement kit that has been used for a long time, we used it as a positive control when evaluating ADCHECK's performance. We added a sentence explaining the purpose of using the BANA test to the `Result’ part.
8.- Page 3, Line 99; Please correct the typo (Name “Loesche”).
We corrected it (L113).
9.- Page 4, Line 131; The authors mentioned that "Genomic DNA was isolated from cultured bacterial cells using the QIAamp DNA Mini Kit". Did you also isolate the bacterial DNA from tongue swab samples? Please add a description how you process the clinical samples for PCR analysis.
Thank you for pointing out our error. We isolated bacterial DNA from tongue coat samples for real time PCR. We have corrected the statement regarding clinical samples for PCR analysis (L145-161).
10.- Page 4, Line 145; Please replace reference 11 with the correct one.
We corrected it.
11.- Page 4, Line 153; Considering the objective of the present study, the TLP activity level of cultured T. f and T. d should be measured.
We agree that additional experiments on T.f and T.g as you suggested would be valuable. In this study, P. g was used as a representative bacterium that produces TLPs such as gingipain. We will investigate this point and intend to report it in a later paper.
12.- Page 6, Line 201; The authors mentioned, "ADCHECK was able to detect the red complex pathogens present in the patient samples” and “the ADCHECK is a test kit that can detect red-complex bacteria stably and with high accuracy (Line 228)”. These sentences do not accurately represent the results and should be revised.
Thank you for your very targeted comments.
As you pointed out, ADCHECK measures TLPs produced by Red Complex bacteria. Based on this fact, we have corrected these statements that were misrepresented.
- Page 7, Line 211; The authors mentioned that "the ADCHECK is a useful tool for the treatment of periodontitis”. This sentence is unclear, and please explain in detail the benefit to use ADCHECK for periodontal treatments.
We have corrected the sentence as you have indicated (L246-250).
Reviewer 2 Report
The work is very interesting, I consider it suitable for publication. However, some adjustments must be made to improve the clarity of the document and better understanding.
Abstract
In the abstract it is not appropriate to use abbreviations, they should be used from the introduction.
Taxonomically it is not appropriate to abbreviate the names of bacteria as they do in the abstract line 17 and 18, “Porphyromonas gingivalis (P. g), Tannerella forsythia (T. f), and Treponema denticola (T. d)” Instead it should be the full name the first time Porphyromonas gingivalis appears and then P. gingivalis in italics.
Introduction
The same comment regarding the nomenclature of the bacteria in the abstract. Paragraph 42 and 43, please clarify if this trypsin-type enzyme that you speak of, which is common in red complex bacteria, corresponds to which enzymes? Gingipains?
Materials and methods
Clinical samples
Justify why you included this number of patients. Did you do any sample size calculations? Or is it a convenience sample? Justify this part.
TLP-AA kit
The authors indicate that they had no funding for this work, but according to what is understood they are validating this kit that they had previously developed, it would be appropriate to clarify this part, was it the same group that previously developed it?
Bacterial cultures
Clarify the incubation time of P. gingivalis
Real-time PCR
Paragraph 136 “20 ml total PCR amplification volume for each reaction were used” to confirm the volume, which is enough to be PCR, or how the methodological strategy was, since it is not clear.
Statistical analysis
How was the statistical analysis of the clinical part?
Results
Table 1. The name of the color column does not look good
Table 2. Please put the bacteria with the universal and adequate nomenclature
The evaluation of the samples that they included is not evidenced in the tables, there is no information on the clinical part and the detection of the bacteria with the kit that they are testing together with the comparison by qPCR, there is only the information on sensitivity and specificity, but There is no clinical comparison between healthy individuals and those with periodontitis. How was the behavior?
Discussion
Clinical data is discussed in the discussion and there is not much explanation of this part in the results.
Conclusion
It should be expanded a little more
Author Response
Reviwer 2
Thank you for your kind suggestions.
We agree with you and have incorporated this suggestion throughout our paper.
Here are the corrections we have made in response to the points you raised
- Abstract
In the abstract it is not appropriate to use abbreviations, they should be used from the introduction.Taxonomically it is not appropriate to abbreviate the names of bacteria as they do in the abstract line 17 and 18, “Porphyromonas gingivalis (P. g), Tannerella forsythia (T. f), and Treponema denticola (T. d)” Instead it should be the full name the first time Porphyromonas gingivalis appears and then P. gingivalis in italics.
We have corrected the use of abbreviations in the abstract and the use of bacterial names throughout the paper.
- Introduction
The same comment regarding the nomenclature of the bacteria in the abstract. Paragraph 42 and 43, please clarify if this trypsin-type enzyme that you speak of, which is common in red complex bacteria, corresponds to which enzymes? Gingipains?
Red complex bacteria produce various types of Trypsin-like proteases, and gingipain produced by P. gingivalis is one of these Trypsin-like proteases. However, they are called Trypsin-like proteases because they do not have names such as gingipain.
- Clinical samples
Justify why you included this number of patients. Did you do any sample size calculations? Or is it a convenience sample? Justify this part.
We determined our sample size in this study according to the following papers. It is not convenience sample. Static power is over 80%.
Brown LD, Cai TT, DasGupta A (2001) Interval Estimation for a Binomial Proportion, Statistical Science , 16:2, 101-117, doi:10.1214/ss/1009213286
Bujang, M. A., & Adnan, T. H. (2016). Requirements for minimum sample size for sensitivity and specificity analysis. Journal of clinical and diagnostic research: JCDR, 10(10), YE01.
- TLP-AA kit
The authors indicate that they had no funding for this work, but according to what is understood they are validating this kit that they had previously developed, it would be appropriate to clarify this part, was it the same group that previously developed it?
As you have indicated, several of the authors of this paper (including myself) have participated in a study using this kit to examine the tongue coating of patients with periodontal disease to verify their susceptibility to periodontal disease (ref 7). However, the samples used in this study are from different patient populations and are separate projects. In addition, we have not received any funding from Kyushu Dental University for this study.
- Bacterial cultures
Clarify the incubation time of P. gingivalis
The incubation time for P. gigivalis is 24 hours. We have added this description to Methods and Materials.
- Real Time PCR
Paragraph 136 “20 ml total PCR amplification volume for each reaction were used” to confirm the volume, which is enough to be PCR, or how the methodological strategy was, since it is not clear.
We have corrected the RT-PCR part of the Materials and Methods as you pointed out
- Statical analysis
How was the statistical analysis of the clinical part?
We have corrected Statical analysis part of the Materials and Methods as you pointed out
Results
- Table 1. The name of the color column does not look good
- Table 2. Please put the bacteria with the universal and adequate nomenclature
We have corrected the above two points (Table 2 (previous Table1): P4, Table 3 (previous Table2): P5).
- The evaluation of the samples that they included is not evidenced in the tables, there is no information on the clinical part and the detection of the bacteria with the kit that they are testing together with the comparison by qPCR, there is only the information on sensitivity and specificity, but There is no clinical comparison between healthy individuals and those with periodontitis. How was the behavior?
We acknowledge the reviewer's comment on this point. The purpose of this study is to compare the number of Red Complex bacteria in patient tongue samples by PCR with the ADCHECK score values. Therefore, we did not compare ADCHECK score values or bacterial counts by RT-PCR with clinical parameters such as periodontal pocket depth. The relationship between AD CHECK score values and periodontitis status was reported in a previous study (ref 8). The characteristics of the healthy subjects and periodontitis patients who provided samples for the current study are described in Table 1.
Discussion
- Clinical data is discussed in the discussion and there is not much explanation of this part in the results.
We are uncertain as to the meaning of the reviewer's comment.
Conclusion
- It should be expanded a little more
In accordance with the reviewer’s comment, we revised conclusion part (L307).
Round 2
Reviewer 1 Report
Thank you for your revision. I would just request some minor corrections.
- Figure 3; Thank you for showing additional information. This reviewer recommends performing a trend test to verify the correlation between ADCHCK score and the number of red complex bacteria. Also, please discuss the result of Figure 3.
- Discussion, Line 241-244; It sounds repetitious, but the descriptions do not accurately represent the results of this study. It is not verified There is not verified that ADCHECK score is not affected by TLPs derived from black pigment anaerobes not included in the red-complex bacteria. ADCHECK is a kit for easy detection of TLPs, and the score was correlated with the number of red complex bacteria in the tongue swab samples, isn’t it? (please revalidate and modify the sentence of underline based on the statistical results of Figure 3)
Please revise the related description in the Conclusion part.
Author Response
Reviewer 1
Thank you for your further suggestions.
We are also sorry that we did not grasp some of your intentions during 1st review and revise. We revised our manuscript further in accordance with your suggestions.
The followings are our response to your comments.
- - Figure 3; Thank you for showing additional information. This reviewer recommends performing a trend test to verify the correlation between ADCHCK score and the number of red complex bacteria. Also, please discuss the result of Figure 3.
We performed a trend test to verify the correlation between ADCHECK score and the number of red complex pathogens (Fig 3b). We found that the there was a correlation between the number of red complex bacteria and the ADCHECK score value (r = 0.580). We have also added a discussion of these results to the Discussion Part (L270-283).
- - Discussion, Line 241-244; It sounds repetitious, but the descriptions do not accurately represent the results of this study. It is not verified There is not verified that ADCHECK score is not affected by TLPs derived from black pigment anaerobes not included in the red-complex bacteria. ADCHECK is a kit for easy detection of TLPs, and the score was correlated with the number of red complex bacteria in the tongue swab samples, isn’t it? (please revalidate and modify the sentence of underline based on the statistical results of Figure 3)
We revised this part in accordance with your suggestions (L251-256).
- Please revise the related description in the Conclusion part.
We revised this part in accordance with your suggestions (L333-337).
Reviewer 2 Report
I thank the authors for having taken into account the observations made, I believe that this has improved the manuscript.
However, let me insist on including in your manuscript the sample size calculation with the references you have in the answers.
Additionally, despite including some clinical data in the table, I do not see the detection of bacteria by qPCR by diagnosis, I think this is important.
In addition to justifying a little the significant difference in age of the population studied, although this is not an epidemiological study, it is important to justify this part and the importance of the samples in the validation of the Kit.
Finally, I ask you to confirm and improve the description of the “Bacterial cultures” section because the growth of P. gingivalis is described and they indicate that it is 24 hours, which seems quite strange to me since this is a bacterium that is difficult to grow that requires at least 72 hours under anaerobic conditions with an adequate source of nutrition, additionally do not describe the growth of the other two bacteria T. forsythia and T. denticola.
And nothing has described this part of the other bacteria that appear in table 2.
Author Response
Reviewer 2
Thanks for your kind and helpful review on our paper.
We have revised our manuscript as much as possible in accordance with your suggestions.
The followings are our responses for your comments.
- However, let me insist on including in your manuscript the sample size calculation with the references you have in the answers.
We have described how we obtained our sample size and the references on which we based it (L175-179). In that reference (ref 13), sample size is determined from a table, so we have not included any formulas.
- Additionally, despite including some clinical data in the table, I do not see the detection of bacteria by qPCR by diagnosis, I think this is important.
According to your suggestion, we have added data on the number of red complex pathogens detected on the tongue coat to the patient information (Table 1).
- In addition to justifying a little the significant difference in age of the population studied, although this is not an epidemiological study, it is important to justify this part and the importance of the samples in the validation of the Kit.
We have added a sentence describing the significant differences between the periodontitis group and the healthy group in accordance with your suggestion.
- Finally, I ask you to confirm and improve the description of the “Bacterial cultures” section because the growth of gingivalis is described and they indicate that it is 24 hours, which seems quite strange to me since this is a bacterium that is difficult to grow that requires at least 72 hours under anaerobic conditions with an adequate source of nutrition, additionally do not describe the growth of the other two bacteria T. forsythia andT. denticola.
We appreciate your comment on this point. As you pointed out, we had previously incorrectly stated the incubation time for P. gingivalis. The correct incubation time for P. gingivalis is 72 hours. The bacterial culture section of Materials and Methods has been revised. Since T. forsythia and T. denticola are not used in this study, their culture methods are not described.
- And nothing has described this part of the other bacteria that appear in table 2.
Thank you for appropriate comments. Table S1 was prepared as supplementary information for the bacterial culture methods used in the ADCHECK cross-resistance test, since there were a very large number of bacteria (40 species) cultured in the study. Table S1 describes the medium, temperature, and incubation period used to culture these bacteria.